# Mitigating backdoor attacks with generative modelling and dataset relabeling

## Abstract

Data-poisoning attacks change a small portion of the training dataset by introducing hand-crafted triggers and rewiring the corresponding labels towards a desired target class. Training on such data injects a backdoor into the model, that causes incorrect inference in selected test examples. Existing defenses mitigate the risks of such attacks through various modifications of the standard discriminative learning procedure. This paper explores a different approach that promises clean models by means of per-class generative modelling. We start by mapping the input data into a suitable latent space by leveraging a pre-trained self-supervised feature extractor. Interestingly, these representations get either preserved or heavily disturbed under recent backdoor attacks. In both cases, we find that per-class generative models give rise to probabilistic densities that allow both to detect the poisoned data and to find their original classes. This allows to patch the poisoned dataset by reverting the original labels and considering the triggers as a kind of augmentation. Our experiments show that training on patched datasets greatly reduces attack success rate and retains the clean accuracy. We aim to publish the source code for reproducing our experiments in the final version of the paper.

## 1 Introduction

Deep neural networks (DNNs) are progressively establishing themselves as the default approach for resolving diverse problems across various domains He et al. (2016); Xiong et al. (2016); Silver et al. (2016). However, their extraordinary capacity leaves them vulnerable to various cybernetic attacks (Biggio & Roli, 2018). One type of those attacks inserts a backdoor to trained models by introducing subtle changes to the training dataset (Mei & Zhu, 2015). We often denote these changes as data poisoning. The installed backdoor appropriates some of the model capacity in order to embed malicious behaviour according to the attacker's goals Laskov & Lippmann (2010). In discriminative models, the malicious behaviour often involves a wrong prediction in presence of a pre-defined trigger Chen et al. (2017). These attacks typically promote the backdoor stealthines by retaining high generalization performance on benign samples Gu et al. (2019).

Data poisoning attacks must trade-off stealthiness with applicability. Localized attacks can be easily applied in the physical world, however they can be uncovered by careful visual inspection (Gu et al., 2019; Chen et al., 2017). Pervasive attacks may be imperceptible to the human eye but they can not be applied with a sticker (Wang et al., 2019; Li et al., 2021c). Many early defenses are effective only on particular kinds of attacks (Wang et al., 2019). Recent defenses aim at broad applicability by avoiding any prior knowledge about the attack (Huang et al., 2022; Chen et al., 2022; Gao et al., 2023). Our approach shares this ambition while bringing forward several novelties.

In this work, we propose a novel approach to prevent the backdoor deployment given a potentially poisoned dataset from an untrusted source Wang et al. (2019). Besides the removal of poisoned samples, our method allows to relabel some of the poisoned samples according to the predictions from a generative classifier Mackowiak et al. (2021). The attack gets neutralized due to changed labels even though the triggers remain in images. Experiments show that this step succeeds to improve the accuracy on clean test data without increasing the attack success rate.

We hypothesize that poisoning attacks give rise to strong semantic difference between the clean and poisoned samples of the target class. Poisoned samples are generated by either **i)** injecting triggers to the images of non-target classes (Gu et al., 2019; Chen et al., 2017; Qi et al., 2022) or **ii)**

applying strong perturbations to the images of the target class (Turner et al., 2019). In both cases, the poisoned samples get displaced outside of the target class manifold. Hence, one could expect that a generative model of the target class should assign these samples a lower likelihood. However, generative modelling of RGB images may behave in an unintuitive manner. Specifically, studies have shown that generative models may assign high likelihoods to outliers (Nalisnick et al., 2018; Serrà et al., 2019). In order to avoid that, we decide to escape from the input space into the latent space by relying on a separately trained self-supervised feature extractor.

We proceed by modelling the probabilistic distribution for each class. We identify potentially poisoned data according to the following two tests. The first test identifies samples residing within the distribution of a class that differs from their label. The second test identifies samples that are outside the distribution of the entire dataset. This step cleanses most of the poisoned samples from the dataset. We attempt to restore the original labels of poisoned samples through generative classification Mackowiak et al. (2021). Finally, we train the deployment model on relabeled data.

To summarize, our contributions encompass three key aspects. (1) We categorize effects of various backdoor attacks to self-supervised image representations. (2) We propose the first backdoor defense based on generative modelling. (3) The proposed method can cleanse the dataset by pseudo-labeling the poisoned samples according to the predictions of our generative classifier. Our experiments demonstrate the effectiveness of our approach in comparison to several state-of-the-art defenses. Importantly, our method successfully defends against a variety of attack types, including the latest advances that claim to undermine defenses based on latent-separability (Qi et al., 2022).

## 2 RELATED WORK

Backdoor attacks are an emerging and rapidly growing research area that poses serious security threats to the usage of DNNs in the community. This paper focuses on the poisoning-based backdoor attacks, where the attacker can only inject poisoned examples into the training set while being unable to modify other components, such as training loss and model architecture. The main goal of existing backdoor attacks is to increase the attack success rate (Li et al., 2022) while retaining stealthy triggers and low poisoning rates (Gu et al., 2019). Crafting inventive triggers can significantly contribute to the attack stealthiness Li et al. (2020). Many kinds of triggers were introduced, including black-white checkerboards (Gu et al., 2019), blending backgrounds (Chen et al., 2017), invisible noise Li et al. (2020), adversarial patterns (Zhao et al., 2020b) and sample-specific patterns (Li et al., 2021c; Nguyen & Tran, 2020). Existing attacks can further be divided into poisoned-label and clean-label types. Poisoned-label approaches (Gu et al., 2019; Chen et al., 2017; Nguyen & Tran, 2021; Li et al., 2021c; Nguyen & Tran, 2020) connect the trigger with the target class by relabeling poisoned samples as target labels. Clean-label approaches affect samples from the target class while leaving the labels unchanged. However, they suffer from lower performance compared to poison-label attacks (Zhang et al., 2020; Li et al., 2022).

Recent work has explored various methods to mitigate the risks of the backdoor-related threats. Existing defenses can be classified into three categories, including i) detection-based defenses (Tran et al., 2018; Chen et al., 2018; Xu et al., 2021; Guo et al., 2023), ii) training-time defenses (Li et al., 2021b;a; Huang et al., 2022; Gao et al., 2023) and iii) post-processing backdoor defenses (Liu et al., 2018; Wang et al., 2019; Dong et al., 2021; Tao et al., 2022; Zhao et al., 2020a). The goal of detection-based defenses is to discover backdoored DNNs or poisoned samples in order to deny their impact. Training-time defenses aim to develop a clean model from a potentially poisoned dataset. Post-processing defenses intend to eliminate or reduce the impact of the backdoor within the already poisoned model. Our defense shares some attributes with detection-based and training-time defenses. Its core revolves around identification and removal of the poisoned samples. Our ultimate objective is to restore the correct labels and to produce a clean model with efficient performance.

The primary drawback of detection-based defenses is the unused potential of the filtered poisoned samples (Chen et al., 2018). On the other hand, training-time defenses Huang et al. (2022); Gao et al. (2023) remain vulnerable to the retained poisoned samples Jia et al. (2022). We address these limitations by proposing a novel approach that detects poisoned samples and corrects their labels through generative inference. Now the triggered data act as data augmentation instead of being an instrument for backdoor deployment. This effectively prevents the model to learn the association between the trigger and the target class.

# 3 MOTIVATION

Data poisoning attacks pose a great challenge since the attackers hold the first-mover advantage (Gu et al., 2019). Nevertheless, we know that the triggers must not disturb the image semantics in order to promote stealthiness (Li et al., 2022). We propose to take advantage of this constraint by grounding our defense on image content while minimizing the influence of poisoned labels.

## 3.1 EMBEDDING THE INPUT DATA INTO THE LATENT SPACE

We consider backdoor defenses that avoid standard supervised learning due to its sensitivity to poisoned labels and susceptibility to overfitting. Consider a poisoned traffic sign dataset where the priority road class contains some triggered instances of the stop sign (Gu et al., 2019). Then, an ideal approach would aim to detect triggered stop signs according to some generative anomaly score.

However, generative models of RGB images learn low-level features that do not correlate with high-level semantics (Kirichenko et al., 2020). Nevertheless, we know that generative modelling may deliver more intuitive performance in semantic latent spaces (Zhang et al., 2020; Blum et al., 2021). Consequently, we embed the input data into the class-agnostic latent space of a popular self-supervised model Chen et al. (2020).

The chosen embedding compresses the input data by eliminating extraneous details and noise, and thus makes it suitable for generative modelling. Experiments confirm our intuition that backdoor attacks affect self-supervised features to a lesser extent than their supervised counterparts. In fact, poisoned self-supervised embeddings remain on the manifolds of their clean class in cases of most recent attacks as we show next.

## 3.2 BEHAVIOUR IN FEATURE SPACE

Recent studies develop their intuitions by visualizing the effects of poisoning within the latent space (Huang et al., 2022; Chen et al., 2022). We apply their methodology to study effects of prominent poisoning attacks to self-supervised embeddings.

Figure 1 (left) shows that most attacks disperse poisoned samples among the clean samples of their original class (Gu et al., 2019; Chen et al., 2017; Nguyen & Tran, 2021; Qi et al., 2022). We refer to this scenario as non-disruptive poisoning since it exerts only a minor influence to self-supervised embeddings. This behaviour is intuitively clear since poisoned samples share more similarities with their original class than with the target class. We remind the reader that the triggers are designed for minimal visual impact in order to conceal the attack.

Figure 1 (right) illustrates the attacks that operate by isolating the poisoned samples from the rest of the data (Turner et al., 2019). The isolation occurs due to influential triggers or strong perturbations of the original images. Consequently, we denote this scenario as disruptive poisoning. A prominent representative of this type is the clean-label attack, which adversarially perturbs samples within the target class in order to make the predictions uncertain (Turner et al., 2019).

## 3.3 MODELING PER-CLASS DISTRIBUTIONS

Both scenarios from Figure 1, displace the poisoned samples far away from the clean samples of the target class. We propose to expose this occurrence by comparing probabilistic per-class densities of the latent embedings. These densities can be recovered by learning per-class generative models such as normalizing flows (Rezende & Mohamed, 2015) or variational encoders (Kingma & Welling, 2014). Moreover, in the non-disruptive scenario, the original class of the poisoned sample can be recovered through generative classification.

# 4 PROPOSED METHOD

## 4.1 PROBLEM FORMULATION

**Threat model.** We adopt the poisoning-based threat model where the attacker poisons a subset of the original benign training dataset $\mathcal{D} = \{(\boldsymbol{x}_i, y_i)\}_{i=1}^{N} \subset \mathcal{X} \times \mathcal{Y}$, where $\mathcal{X}$ contains all possible

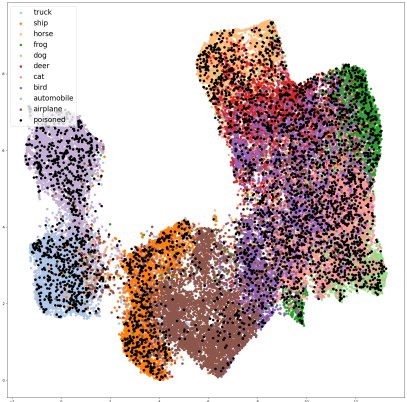 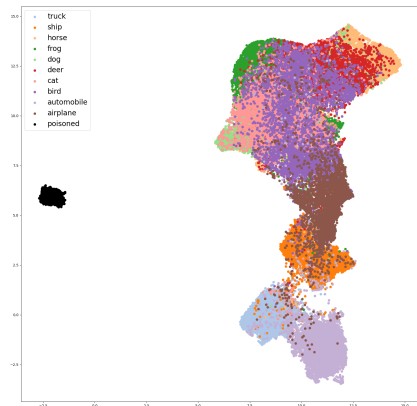

Figure 1: 2D UMAP visualization of clean and poisoned samples in the self-supervised feature space for CIFAR-10. Poisoned samples are shown in black, clean targets are in brown, while clean samples from other classes are shown in other colors. Non-disruptive attacks (left, Gu et al. (2019)) displace the samples within the borders of their original manifold. Disruptive attacks (right, Turner et al. (2019)) make the poisoned samples completely distinct from the rest of the dataset.

inputs and $\mathcal{Y}$ is the set of classes (Gao et al., 2023; Huang et al., 2022). The attack specifies a single target label $y_\mathrm{T} \in \mathcal{Y}$ and an operation $\oplus$ that adds a trigger $\boldsymbol{\delta}$ to an input. Hence, the poisoned subset is $\mathcal{D}_\mathrm{P} = \{(\bar{\boldsymbol{x}}_i, y_\mathrm{T})\}_{i \in \mathcal{I}_\mathrm{P}}$, where $\tilde{\boldsymbol{x}}_i = \boldsymbol{x}_i \oplus \boldsymbol{\delta}$ and $\mathcal{I}_\mathrm{P}$ is the set of poisoned indices. Note that $y_\mathrm{T}$ may equal the clean label $y_i$ (Turner et al., 2019). The remaining samples from $\mathcal{D}$ form the clean subset $\mathcal{D}_\mathrm{C}$, which is combined with $\mathcal{D}_\mathrm{P}$ to form the poisoned dataset $\tilde{\mathcal{D}} = \mathcal{D}_\mathrm{C} \cup \mathcal{D}_\mathrm{P}$. The aim of the attack is for a model trained on $\tilde{\mathcal{D}}$ to classify triggered test inputs into the target class $y_\mathrm{T}$ without affecting the performance on clean inputs.

**Defender's goals.** We assume that the defender completely controls the training process. Given a possibly poisoned training set $\tilde{\mathcal{D}}$, the defender's objective is to obtain a trained model instance without a backdoor while preserving high accuracy on benign samples. Our defense aims to achieve this goal by transforming $\tilde{\mathcal{D}}$ into a cleansed dataset $\hat{\mathcal{D}}$ that allows safe training according to the standard training procedure.

## 4.2 DEFENSE OVERVIEW

The input to our method is a potentially poisoned dataset, denoted as $\tilde{\mathcal{D}}$. Figure 2 shows that our method consists of two stages. The first stage starts by training a self-supervised feature extractor $f_{\boldsymbol{\theta}_\mathrm{F}} \colon \boldsymbol{x} \mapsto \boldsymbol{z}$ on $\tilde{\mathcal{D}}$. Then, it trains per-class generative models $p_{\boldsymbol{\theta}_y} \colon \boldsymbol{z} \mapsto p(\boldsymbol{z} \mid y, \boldsymbol{\theta}_\mathrm{g})$ on the features $\{f_{\boldsymbol{\theta}_\mathrm{F}}(\boldsymbol{x}_i) \mid y_i = y\}$ for each class $y \in \mathcal{Y}$. We hypothesize inductive biases such that the model for class $y$ assigns high densities to poisoned samples of the original class $y$. Each class with a considerable number of low density samples can be considered a potential target class.

Once we identify the target classes, we separate their samples into three distinct categories. The clean category $\hat{\mathcal{D}}_\mathrm{C}$ contains samples of the target classes with high density of the label class and low density of other classes. The poisoned category $\hat{\mathcal{D}}_\mathrm{P}$ contains samples with high density of non-label classes and lower density of the label class. Finally, the category $\hat{\mathcal{D}}_\mathrm{O}$ contains ambiguous samples with uncertain poisoning status and class membership.

We can safely reintegrate clean samples $\hat{\mathcal{D}}_\mathrm{C}$ back into the cleansed dataset $\hat{\mathcal{D}}$. We reject samples from $\hat{\mathcal{D}}_\mathrm{O}$ and reintegrate samples from $\hat{\mathcal{D}}_\mathrm{P}$ into $\hat{\mathcal{D}}$ after relabeling them with the class exhibiting the highest density. The dataset $\hat{\mathcal{D}}$ is the outcome of our defense and it is sent back to the retraining process.

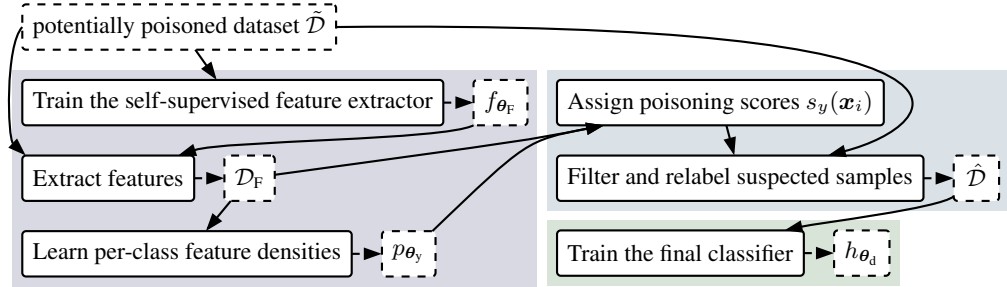

Figure 2: Overview of the proposed defense. $\mathcal{D}_F$ denotes the set of all features $\{f_{\boldsymbol{\theta}_F}(\boldsymbol{x}_i)\}$.

.

### 4.3 MODELING PER-CLASS DENSITIES OF SELF-SUPERVISED FEATURES

We learn a self-supervised feature extractor $f_{\boldsymbol{\theta}_F}$ by applying SimCLR (Chen et al., 2020) to $\tilde{\mathcal{D}}$. Then, we learn class-conditional densities of features $\boldsymbol{z}_i = f_{\boldsymbol{\theta}_F}(\boldsymbol{x}_i)$ with lightweight per-class normalizing flows.

A normalizing flow is a bijective mapping $g_{\boldsymbol{\theta}_{NF}}$ that transforms the input $\underline{\boldsymbol{z}}$ with a complex distribution into an output $\underline{\boldsymbol{u}}$ with a fixed simple distribution. The simple distribution is usually chosen to be an isotropic Gaussian with a zero mean and unit variance: $g_{\boldsymbol{\theta}_{NF}}(\boldsymbol{z}) = \underline{\boldsymbol{u}} \sim \mathcal{N}(\boldsymbol{0}_d, \mathbf{I}_d)$, where $d$ is the dimension of the input. Assuming the bijective mapping has the necessary properties, the density of the inputs can be computed by applying change of variables:

$$p(\boldsymbol{z}) = p(\boldsymbol{u})\left|\det \frac{\partial \boldsymbol{u}}{\partial \boldsymbol{z}}\right| \tag{1}$$

A normalizing flow is usually implemented as a sequence of simpler invertible mappings with learnable parameters, such as affine coupling layers (Dinh et al., 2016).

We define a simple class-conditional model as a set of normalizing flows with separate parameters $\boldsymbol{\theta}_y$ for each class. We express the class-conditional density as $p_{\boldsymbol{\theta}_y}(\boldsymbol{z}) = p(\boldsymbol{z} \mid y, \boldsymbol{\theta}_g)$, where $\boldsymbol{\theta}_y = \boldsymbol{\theta}_{g_{[y]}}$ are the parameters of the model instance that corresponds to class $y$. Let $\mathcal{D}_F^y = \{f_{\boldsymbol{\theta}_F}(\boldsymbol{x}_i) \mid y_i = y\}$. We train the generative models by minimizing the average negative log-likelihood:

$$\overline{\mathcal{L}}(\boldsymbol{\theta}_y, \mathcal{D}_F^y) = \mathop{\mathbb{E}}_{(\boldsymbol{z},y) \in \mathcal{D}_F^y} -\log p_{\boldsymbol{\theta}_y}(\boldsymbol{z}). \tag{2}$$

Each normalizing flow consists of a pair of affine coupling modules. Each coupling layer contains two learnable affine operations with a ReLU activation between them.

After estimating the densities, our next objective is to determine whether the poisoning is disruptive and to identify the classes containing poisoned samples.

### 4.4 IDENTIFYING TARGET CLASSES

We first check for the presence of non-disruptive poisoning by assuming that the poisoned samples resemble their source classes in the self-supervised feature space. In this case, the generative model of the target class will assign moderate densities to many foreign examples due to learning on triggered examples. This behaviour will be much less pronounced in non-target classes since the triggered examples represent a minority of the whole dataset. Consequently, we propose to identify target classes according to the average log-density over all foreign examples.

$$S_{ND}^y = \overline{\mathcal{L}}(\boldsymbol{\theta}_y, \{f_{\boldsymbol{\theta}_F}(\boldsymbol{x}_i) \mid y_i \neq y\}). \tag{3}$$

We classify a class $y$ as non-disruptively poisoned if $S_{ND}^y$ is lower than the threshold $\beta_{ND}$.

Next we search for disruptive poisoning. The defining characteristic of this type of poisoning, is that the poisoned samples are much more dissimilar from the clean samples than the clean samples

of different classes among themselves. As a consequence, we expect the foreign densities in such samples to be much lower than the foreign densities in the clean samples. Our approach searches for classes with a significant number of such outliers. We formalize this idea by first defining the maximum foreign density score for each example:

$$v_y(\boldsymbol{z}) = \max_{y' \in \mathcal{Y}, y' \neq y} p_{\boldsymbol{\theta}_{y'}}(\boldsymbol{z}) \tag{4}$$

We classify a class as disruptively poisoned if the fraction of samples with low $v$ scores (4) exceeds the threshold $\beta_\mathrm{D}$. More precisely, for each class $y$, we i) compute the set of v scores of the corresponding samples $\mathcal{V}_y = \{v_{y_i}(\boldsymbol{z}_i) \mid y_i = y\}$, ii) compute a histogram with 30 bins of equal widths for $\mathcal{V}_y$ as shown in Figure 3, iii) find the value $\mu_y$ of the minimum of the histogram on the left from the hyperparameter $\lambda$, and (iv) compute the fraction of examples with $v_y(\boldsymbol{z}_i) < \mu_y$:

$$S_\mathrm{D}^y = \frac{|\{\boldsymbol{z}_i : y_i = y, v_y(\boldsymbol{z}_i) < \mu_y\}|}{|\{\boldsymbol{z}_i : y_i = y\}|} \tag{5}$$

Finally, we classify a class $y$ as disruptively poisoned if $S_\mathrm{D}^y$ is less than the threshold $\beta_\mathrm{D}$. We can interpret $\beta_\mathrm{D}$ as the minimum fraction of poisoned samples per class. Too low values of $\beta_\mathrm{D}$ can result in false positives, while too high values can result in false negatives when the fraction of poisoned samples is smaller.

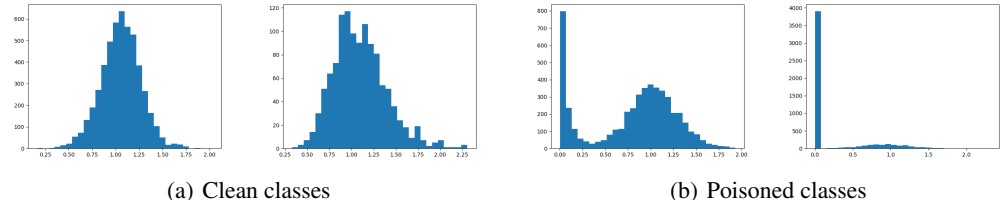

(a) Clean classes          (b) Poisoned classes

Figure 3: The histograms illustrate the distribution of the maximum foreign density scores $v_y(\boldsymbol{z})$ of clean and poisoned classes. In contrast to clean classes, the poisoned classes exhibit strong bimodality. On the right, the first histogram refers to the poisoned CIFAR-10 class with the LC attack (Turner et al., 2019), and other to the ImageNet class poisoned with the BadNets attack (Gu et al., 2019).

### 4.5 IDENTIFYING POISONED EXAMPLES

After identifying the poisoned classes, this steps extracts clean samples $\mathcal{D}_\mathrm{C}$ and poisoned samples $\hat{\mathcal{D}}_\mathrm{P}$. We identify poisoned examples by computing the poisoning score for each example from every class $y$ classified as poisoned:

$$s_y(\boldsymbol{z}) = \frac{p_{\boldsymbol{\theta}_y}(\boldsymbol{z})}{\max_{y' \in \mathcal{Y}, y' \neq y} p_{\boldsymbol{\theta}_{y'}}(\boldsymbol{z})} \tag{6}$$

The poisoning score assumes that, on average, the numerator $p_{\boldsymbol{\theta}_y}(\boldsymbol{z})$ will be higher for clean samples than for poisoned ones. In the case of non-disruptive poisoning, we expect the denominator to be relatively high due to similarity with other samples from its original class. Consequently, clean samples score higher than the poisoned ones. In disruptive poisoning, the numerator will be extremely low because the poisoned samples are entirely outside the distributions of all clean classes. Consequently, in this case, the poisoned samples will score significantly higher than clean samples.

### 4.6 FILTERING AND RELABELING OF POISONED EXAMPLES

We split the samples from identified target classes into three parts according to the hyperparameter $\alpha \in (0..0.5]$. We include the samples with $\alpha$ highest poisoning scores in $\hat{\mathcal{D}}_\mathrm{P}$, and include the samples

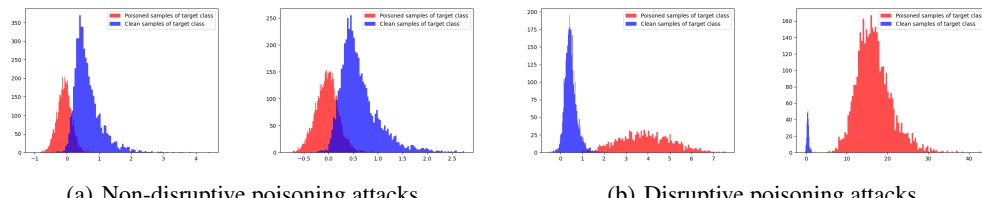

(a) Non-disruptive poisoning attacks         (b) Disruptive poisoning attacks

Figure 4: The values of the score metric for all samples within one target class. Clean samples are shown in blue, poisoned in red. Results displayed on the left correspond to the BadNets and Blend attacks on CIFAR-10, while results shown on the right are for the Label-Consistent attack on CIFAR-10 and BadNets attack on ImageNet.

with the $\alpha$ lowest poisoning scores together with samples from identified clean classes in $\hat{\mathcal{D}}_{\mathrm{C}}$, and ignore the $1 - 2\alpha$ of intermediate samples with uncertain poisoning scores.

Finally, the clean and relabeled samples are combined into the cleansed dataset:

$$\hat{\mathcal{D}} = \hat{\mathcal{D}}_{\mathrm{C}} \cup \{(\boldsymbol{x}, \operatorname*{arg\,max}_{y' \in \mathcal{Y}, y' \neq y} p_{\boldsymbol{\theta}_y}(f_{\boldsymbol{\theta}_{\mathrm{E}}(\boldsymbol{x})})) \mid (x, y) \in \hat{\mathcal{D}}_{\mathrm{P}}\} \tag{7}$$

Then, we can train a classifier on the $\hat{\mathcal{D}}$ using the standard training procedure.

## 5 EXPERIMENTS

### 5.1 EXPERIMENTAL SETUPS

**Datasets and DNNs** We evaluate our defenses on three datasets: CIFAR-10 (Krizhevsky et al., 2009), the standard ImageNet subset (Deng et al., 2009), and the standard VGGFace2 subset (Cao et al., 2018). We use ResNet18 (He et al., 2016) on CIFAR-10 and Imagenet, and DenseNet-121 (Huang et al., 2017) on VGGFace2. We provide additional training details in Appendix B.

**Attack baselines and configurations** We consider the fliowing 6 baselines: BadNets (Gu et al., 2019), Blend backdoor attack (Blend) Chen et al. (2017), Warping based backdoor attack (WaNet) (Nguyen & Tran, 2021), Sample specific backdoor attack (ISSBA) (Li et al., 2021c), clean-label attack with adversarial perturbations (LC) (Turner et al., 2019) and a recent attack claiming to beat defenses based on latent separability (Adap-Patch and Adap-Blend) (Qi et al., 2022). By choosing those baselines, we cover visible patch-based attacks (BadNets), invisible attacks (WaNet and Blend), sample0specific attacks (ISSBA) and clean-label attacks (LC). We set the target label as $0$ ($y_t = 0$), and the poisoning rate to $10\%$ in poison-label attacks, and $25\%$ in the clean-label attack. We omit some attacks on ImageNet and VGGFace2 since we were not able to reproduce the performance from their papers. We provide detailed per-attack configurations in Appendix B.

**Defense baselines and configurations** We compare our method with 4 state-of-the-art defenses: Neural attention distillation (NAD) (Li et al., 2021b), Anti backdoor learning (ABL) (Li et al., 2021a), Decoupling based defense (DBD) (Huang et al., 2022) and Backdoor defense via adaptive splitting (ASD) (Gao et al., 2023). Due to their lower complexity, we perform hyperparameter search for NAD and ABL baselines. We implement DBD and ASD defenses by leveraging the official implementation (Huang et al., 2022; Gao et al., 2023) and original hyperparameters. It is also worth mentioning that NAD and ASD require a small subset of clean data for each class.

Our defense uses the ResNet-18 backbone (He et al., 2016) for self-supervised features. We use Adam with $(\beta_1, \beta_2) = (0.9, 0.99)$, lr $= 3 \cdot 10^{-4}$. and batch_size $= 256$ on all datasets. We train each normalizing flow for 50 epochs with Adam, $(\beta_1, \beta_2) = (0.9, 0.99)$, lr $= 10^{-3}$ and batch_size $= 16$. We set $\beta_{\mathrm{ND}} = 0.6$, $\beta_{\mathrm{D}} = 0.05$, $\lambda = 0.75$ and $\alpha = 0.15$ based on early validation experiments.

**Evaluation metrics** We adopt two commonly used metrics to evaluate the defense performance, including the accuracy on the clean test dataset (ACC) and the accuracy of predicting poisoned samples as the target label, referred to as attack success rate (ASR).

## 5.2 MAIN RESULTS

Table 1 evaluates effectiveness of our defense under state-of-the-art attacks and compare it with the state-of-the-art defenses. Our defense consistently achieves significantly lower average ASR than the alternatives, with ASR falling below 1% in the majority of assays. We concurrently maintain consistently high ACC accross all datasets. Note that ASD exhibits higher ACC on CIFAR-10 and VGGFace2, but often at the expense of significantly higher ASR. Note also that ASD requires a small number of clean samples of each class, whereas our defense operates without such requirement.

We especially emphasize the superior performance of our defense under Adap-Patch and Adap-Blend attacks. In spite of the intention of these attacks to suppress the latent separation between poisoned samples and the rest of the dataset, self-supervised representations of the poisoned samples still exhibit semantic similarity with their original classes. For that reason, our method successfully defends against such attacks. Conversely, when we examine the outcomes of the ABL, DBD, and ASD defenses against the Adap-Blend attack, we observe a counterintuitive trend: the resulting ASR becomes even higher than without the defense.

Table 1: Comparisons of our proposed method with 4 backdoor-removal defenses on CIFAR-10, Imagenet and VGGFace2 datasets. We mark defenses requiring extra clean data with *. We report the clean test accuracy as ACC (%) and the attack success rate ASR (%). The best defense method, marked in **bold**, should exhibit highest ACC - ASR.

| Dataset ↓ | Defense → | No Defense | | NAD* | | ABL | | DBD | | ASD* | | Ours | |
|---|---|---|---|---|---|---|---|---|---|---|---|---|---|
| | Attack ↓ | ACC | ASR | ACC | ASR | ACC | ASR | ACC | ASR | ACC | ASR | ACC | ASR |
| CIFAR10 | BadNets | 94.9 | 100 | 88.2 | 4.60 | **93.8** | **1.10** | 92.4 | 0.96 | 92.1 | 3.00 | 92.6 | 0.20 |
| | Blend | 94.2 | 98.25 | 85.8 | 3.40 | 91.9 | 1.60 | 92.2 | 1.73 | **93.4** | **1.00** | 92.4 | 2.00 |
| | WaNet | 94.3 | 98.00 | 71.3 | 6.70 | 84.1 | 2.20 | 91.2 | 0.39 | **93.3** | **1.20** | 92.0 | 0.28 |
| | LC | 94.9 | 99.33 | 86.4 | 9.50 | 86.6 | 1.30 | 89.7 | 0.01 | 93.1 | 0.90 | **92.8** | **0.06** |
| | ISSBA | 94.5 | 100 | 90.7 | 0.64 | 89.2 | 1.20 | 83.2 | 0.53 | 92.4 | 2.13 | **90.9** | **0.11** |
| | Adap-Patch | 95.2 | 80.9 | 91.1 | 2.96 | 81.9 | 0.00 | **92.9** | **1.77** | 93.6 | 100 | 90.6 | 0.13 |
| | Adap-Blend | 95.0 | 64.9 | 88.3 | 2.11 | 91.5 | 81.93 | 90.1 | 99.97 | 94.0 | 93.90 | **90.5** | **0.00** |
| | Average | - | - | 85.6 | 4.27 | 88.4 | 12.76 | 90.2 | 15.05 | 93.1 | 28.87 | **91.7** | **0.40** |
| ImageNet | BadNets | 95.3 | 99.98 | 92.7 | 0.42 | 94.3 | 0.24 | 91.2 | 0.54 | 90.7 | 9.72 | **94.8** | **0.00** |
| | Blend | 93.7 | 99.93 | 90.0 | 0.51 | 93.1 | 0.14 | 90.3 | 0.58 | 89.9 | 2.07 | **93.5** | **0.45** |
| | WaNet | 93.5 | 100 | 90.7 | 0.56 | 92.0 | 1.33 | 90.5 | 0.48 | 88.8 | 2.89 | **93.4** | **1.33** |
| | Average | - | - | 91.1 | 0.50 | 93.1 | 1.71 | 90.7 | 0.53 | 89.8 | 4.89 | **93.9** | **0.59** |
| VGGFace2 | BadNets | 93.2 | 100 | 56.1 | 6.50 | 93.9 | 2.67 | 90.3 | 0.00 | 96.2 | 99.1 | **95.2** | **0.09** |
| | Blend | 92.8 | 99.95 | 50.8 | 7.30 | 93.4 | 5.40 | 90.2 | 0.04 | **96.0** | **0.18** | 94.2 | 0.42 |
| | WaNet | 93.7 | 99.60 | 50.4 | 4.20 | 93.2 | 1.48 | 87.2 | 0.00 | 96.9 | 96.5 | **94.9** | **0.14** |
| | Average | - | - | 52.4 | 6.00 | 93.5 | 3.18 | 89.2 | 0.01 | 96.4 | 65.26 | **94.8** | **0.22** |

Table 2 reports relabeling accuracies of our generative classifier. We measure these accuracies by evaluating our generative classifier against the original labels, as they were prior to poisoning. We observe the lowest accuracy in cases of Adap-Patch and Adap-Blend attacks. We attribute this discrepancy to the tendency of these attacks to enhance the similarity between the clean and poisoned samples within the latent space. Consequently, during the relabeling process, certain clean samples may become entangled with the poisoned samples, leading to decreased classification accuracy.

Table 2: Relabeling accuracies (%) of non-disruptive attacks. As stated, the relabeling in our method only occurs if the attack is classified as non-disruptive.

| Attack → Dataset ↓ | BadNets | Blend | WaNet | ISSBA | Adap-Patch | Adap-Blend |
|---|---|---|---|---|---|---|
| CIFAR-10 | 92.6 | 88.3 | 90.3 | 92.6 | 42.1 | 36.5 |

## 5.3 ABLATION STUDIES

**Importance of self-supervision.** Table 3 validates the choice of the pre-trained feature extractor. We compare our choice of SimCLR with supervised training and CLIP, a feature extractor pre-trained on an extremely large dataset of images and the corresponding captions (Radford et al., 2021). We can see that SimCLR and CLIP deliver similar overall performance, while greatly outperforming the supervised representations. This improvement occurs since self-supervised learning is less affected by triggers and not affected by target labels. Interestingly, our defense still works well and maintains high accuracy against the WaNet attack even with supervised representations. Additionally, we notice that CLIP performs quite well. This suggests that the computationally expensive pre-training on the poisoned dataset may not be necessary. Instead, using pre-trained feature extractors like CLIP can be effective. However, the assumption of obtaining a clean pre-trained feature extractor, such as CLIP, may face challenges, as recent research shows how similar models can be backdoored (Carlini & Terzis, 2021; Carlini et al., 2023).

Table 3: Comparison of different approaches for feature extractor learning on CIFAR10

| Attack → | BadNets | | Blend | | WaNet | |
|---|---|---|---|---|---|---|
| Feature extractor ↓ | ACC | ASR | ACC | ASR | ACC | ASR |
| Supervised | 75.1 | 6.00 | 76.7 | 76.4 | 92.4 | 0.42 |
| Self-supervised (ours) | **92.6** | **0.20** | 92.4 | 2.00 | 92.0 | 0.28 |
| CLIP | 93.1 | 0.85 | **93.3** | **1.90** | **92.6** | **0.33** |

**Robustness to different poisoning rates.** Table 4 validates the robustness of our defense under different poisoning rates. We observe a slight increase in ASR for Blend and WaNet attacks with higher poisoning rates. We attribute that to the larger proportion of poisoned samples in the target class, making it harder for our score $s$ to filter out the completely clean subset. While we contend that poisoning attacks should ideally maintain low poisoning rates to enhance their stealthiness, we acknowledge this as one of the limitations of our defense approach.

Table 4: Resistance of our defense to different poisoning rates on CIFAR10

| Poisoning rate ↓ | Attack → | BadNets | | Blend | | WaNet | |
|---|---|---|---|---|---|---|---|
| | Defense ↓ | ACC | ASR | ACC | ASR | ACC | ASR |
| 1% | No Defense | 95.2 | 99.96 | 95.2 | 94.5 | 94.7 | 60.05 |
| | Ours | 90.3 | 0.04 | 89.7 | 0.01 | 90.3 | 0.03 |
| 5% | No Defense | 94.5 | 100 | 94.7 | 99.3 | 94.4 | 95.7 |
| | Ours | 91.5 | 0.26 | 90.8 | 0.10 | 91.6 | 0.1 |
| 10% | No Defense | 95.0 | 100 | 94.6 | 99.69 | 94.5 | 99.0 |
| | Ours | 92.6 | 0.20 | 92.4 | 2.00 | 92.0 | 0.28 |
| 15% | No Defense | 94.5 | 100 | 94.3 | 99.92 | 94.2 | 99.7 |
| | Ours | 92.6 | 0.60 | 82.6 | 3.60 | 91.7 | 8.9 |
| 20% | No Defense | 94.5 | 100 | 94.4 | 99.90 | 94.1 | 98.83 |
| | Ours | 92.4 | 1.20 | 92.3 | 20.30 | 92.2 | 14.5 |

## 6 CONCLUSION

We have presented a novel analysis of the effects of backdoor attacks on self-supervised image representations. Results of that analysis inspired us to propose a novel backdoor defense that allows to detect poisoned classes and samples, as well as to alleviate the damage through relabeling. Extensive evaluation against the state-of-the-art reveals competitive performance. In particular, we note extremely effective ASR reduction of latent separability attacks Adap-Patch and Adap-Blend. We hope that our method can contribute as a tool for increasing the robustness of deep learning applications. Our future work will attempt to further improve the success of generative relabeling.

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

# A IMPLEMENTATION DETAILS

## A.1 DATASETS AND DNNS

We show the details for each dataset used in our evaluations in Table 5. We opted to randomly select 30 classes from the ImageNet and VGGFace2 datasets to create a subset, primarily to address computational time and cost constraints.

Table 5: Summary of datasets and DNN models used in our experiments.

| Dataset | # Input size | # Classes | # Training images | # Testing images | Models |
|---------|-------------|-----------|-------------------|------------------|--------|
| CIFAR-10 | $3 \times 32 \times 32$ | 10 | 50000 | 10000 | ResNet-18 |
| ImageNet | $3 \times 224 \times 224$ | 30 | 39000 | 3000 | ResNet-18 |
| VGGFace2 | $3 \times 224 \times 224$ | 30 | 9000 | 2250 | DenseNet-121 |

# B ATTACK CONFIGURATIONS

**Training setups** We perform backdoor attacks on CIFAR-10 (Krizhevsky et al., 2009) for 200 epochs, adopting the SGD (Zhang, 2004) optimizer with the learning rate set to 0.1, momentum to 0.9 and weight decay to 0.0005. We divide the learning rate by 10 at epochs 100 and 150. We perform random resized crop, random horizontal flip and normalization as data augmentations. On ImageNet (Deng et al., 2009) and VGGFace2 (Cao et al., 2018) we train for 90 epochs. All images are resized to $224x224x3$ before the trigger injection. The other training configurations are same as in CIFAR-10 trainig. We set batch size to 128 in all experiments.

**BadNets** To perform BadNets attacks, we follow the configurations of Gu et al. (2019); Huang et al. (2022); Gao et al. (2023). On CIFAR-10, the trigger pattern is $2x2$ square in the upper left corner of the image. On ImageNet and VGGFace2, we opt for $32x32$ apple logo also placed in the upper left corner.

**Blend** Following (Chen et al., 2017; Huang et al., 2022; Gao et al., 2023), we use "Hello Kitty" pattern on CIFAR-10 and random noise patterns on ImageNet and VGGFace2. Blending ratio on all datasets is set to 0.1.

**WaNet** Although WaNet (Nguyen & Tran, 2021) belongs to the training time attacks, we follow Huang et al. (2022); Gao et al. (2023) to use the warping-based operation to directly generate the trigger pattern. The operation hyperparameters are the same as in Gao et al. (2023).

**Label Consistent** Following Turner et al. (2019), we use projected gradient descent (Madry et al., 2017) to generate the adversarial perturbations within $l_\infty$ ball. Maximum magnitude $\eta$ is set to 16, step size to 1.5 and perturbation steps to 30. Trigger pattern is $3x3$ grid pattern in each corner of the image

**ISSBA** We replicate the ISSBA (Li et al., 2021c) attack by training the encoder model for 20 epochs with secret size 20. We then leverage the pre-trained encoder to generate the poisoned dataset.

**Adap-Patch and Adap-Blend** To replicate these attacks, we search for the combination of cover and poison rate giving the best ASR, while keeping in mind that those rates should not be too high for attack to remain stealthy, as stated in Qi et al. (2022). We set poison and cover rate both to 0.01. Trigger patterns used are the same as in Qi et al. (2022).

## C  DEFENSE CONFIGURATIONS

**NAD**  We implement this method based on the open source code [1]. We find NAD Li et al. (2021b) to be sensitive to its hyperparameter $\beta$, which is why we perform hyperparameter search for the best results among values $\beta \in \{500, 1000, 1500, 2000, 5000\}$.

**ABL**  To reproduce ABL experiments, we also refer to the open source library [2]. We first poison the model for 20 epochs, followed by the process of backdoor isolation which takes 70 epochs. Lastly, we unlearn the backdoor for 5 epochs on CIFAR-10 and ImageNet, and for 20 epochs to VGGFace2. We search for the best results given hyperparameter $\gamma \in \{0, 0.2, 0.4\}$.

**DBD**  In order to reproduce DBD (Huang et al., 2022), we use the official implementation [3]. We follow all configurations as clarified in Huang et al. (2022).

**ASD**  By following the official implementation [4], we reproduce the ASD (Gao et al., 2023). Similar as for DBD, we follow all defense configurations from Gao et al. (2023)

---

[1] https://github.com/THUYimingLi/BackdoorBox
[2] https://github.com/THUYimingLi/BackdoorBox
[3] https://github.com/SCLBD/DBD
[4] https://github.com/KuofengGao/ASD

