# OpenReview forum: "Mitigating backdoor attacks with generative modelling and dataset relabelling"
_ICLR.cc/2024/Conference — Submitted to ICLR 2024_

### Official Review · Reviewer_y1uP · 2023-10-27

**Soundness:** 1 poor
**Presentation:** 2 fair
**Contribution:** 1 poor
**Rating:** 3
**Confidence:** 4

**Summary:**

The authors propose a sanitization-based defense against backdoor attacks.  Specifically, the authors learn generative models over class conditional feature space representations.  These learned distributions are then used to filter suspect training data, with the final model trained over the sanitized dataset.

**Strengths:**

Machine learning models are brittle, and as models are deployed in settings critical to human well-being, model failures can lead to real-world harm. This paper proposes a simple, intuitive method to improve model robustness. The paper follows the classic paradigm of using generative models to improve discriminative models' performance (i.e., robustness).

**Weaknesses:**

Like all other empirical defenses, the authors' method comes with no guarantees on its effectiveness.  In my view, empirical defense papers must meet two necessary conditions to be fit for publication.  Unfortunately, this paper does not meet either.
1. The paper should explicitly note that their method comes with no guarantees and contrast this weakness against the plethora of papers (e.g., [1]) that provide methods certifiably robust to training data attacks.
2. The paper should ideally evaluate against an adaptive attacker who is aware of their defense and actively tries to avoid it.  At the very minimum, the paper should include a convincing discussion of why an adaptive attacker is not feasible or reasonable.

The authors define $D^y_{F}=\\{f_{\theta_F}(x_i) \vert y_i = y \\}$. As I understand it, $D^y_F$ contains the set of features for all training instances whose **true label** is $y$.  (Note at the bottom of page 3, I believe $y_i$ is defined as the true labels).  Of course, $y_i$ for the poisoned data is unknown (otherwise, the problem is trivial).  I cannot determine whether there is a problem with the notation or method, but it seems ${D}_F^y$ is not known as defined.  Perhaps the authors are assuming a clean validation set to learn these generative models (as in other work), but I do not see a discussion of that.  This is a major concern and one reason I rate the soundness as 1.

In the "Questions" section below, I detail a concern about overstating the paper's novel contributions.  I will wait for the authors' response before categorically defining it as a weakness.

I **strongly** recommend either removing or redesigning Figure 2. The flow of the figure is very non-linear and non-intuitive. Best I can determine, the figure could have a linear progression starting at the initial poisoned dataset and terminating with the final trained model. Perhaps the authors chose this non-linear progression to save space, but I would view this as an especially poor choice.
* One potential way to solve this problem entirely is to change Figure 2 to an algorithm.

Several typographical issues exist in the paper.  I provide a non-exhaustive list below. These did not affect my overall score.
* Page 1: "...stealthines..."
* The authors repeatedly use `\citet{...}` in place of `\citep{...}`. See for example the two citations on page 1 in the paragraph that begins "In this work, ...".
* Page 7: "...fllowing...:
* Page 7: ", Sample specific" ("S" should not be capitalized here)
* Page 7: "sample0specific"
* Page 13: In multiple places, the letter "x" is used in math mode when specifying dimensions resulting in the "x" being italics.  Either do not place the x in math mode or better use `\times` instead of "x".

Table 4 should show the minimum poisoning rate where either the attacks or the defenses start to fail.  For example, does your defense still work at 0.1% poisoning rate.
* Poisoning rate is also only one dimension of an attack's strength. Perturbation strength is an orthogonal dimension of attack strength against which the authors' defense is surely highly susceptible but is not explored in the empirical evaluation.

The empirical evaluation's main results use either a 10% or 25% poisoning rate.  In my view, those attack rates are wholly unrealistic for any marginally plausible real-world scenario.  I would go so far as to consider those poisoning rates not meaningful to study since I cannot see a case where an attacker is inserting 25% poisoned data.

The proposed method is studied only in the vision context. Other modalities are not explored or discussed.

### References

[1] Levine et al. "Deep Partition Aggregation: Provable Defenses Against General Poisoning Attacks" ICLR'20201.

**Questions:**

On page 2, you summarize the paper's second contribution writing, "*We propose the first backdoor defense based on generative modeling*."  This is a very broad claim that I suspect is not true. For example, [1] uses backdoor modeling for a backdoor defense back at NeurIPS 2019.   Please speak more to the basis of this claim.

At the beginning of Section 3.1, the authors write, "*We consider backdoor defenses that avoid standard supervised learning due to its sensitivity to poisoned labels and susceptibility to overfitting*." When I encountered this sentence when reading through the paper the first time, I did not understand what the authors meant, and after completing the paper, I am not sure I understood it.

### References

[1] Qiao et al. "Defending Neural Backdoors via Generative Distribution Modeling" NeurIPS'2019.

**Details Of Ethics Concerns:**

This paper proposes a defense against backdoor attacks. Better defenses can always theoretically facilitate better attacks in the future, but that risk is unavoidable and in this case remote.

---

> ### Author Response · Authors · 2023-11-21
>
> **Weaknesses:**
>
> >The authors define $D^y_{F}=\{f_{\theta_F}(x_i) \vert y_i = y \}$. As I understand it, $D^y_F$ contains the set of features for all training instances whose **true label** is $y$. [...]
>
> We apologize, we have made an error in the definition of $D^y_\text{F}$ by omitting $(x_i,y_i) \in \widetilde{\mathcal D}$.
> The original labels are not known and the definition should have been $D^y_\text{F}=\{f_{\theta_F}(x_i) \mid (x_i,y_i) \in \widetilde{\mathcal D}, y_i = y \}$.
>
> >I **strongly** recommend either removing or redesigning Figure 2. [...]
> >One potential way to solve this problem entirely is to change Figure 2 to an algorithm.
>
> Would you consider a description like this as an improvement over Figure 2?
>
> 1. Train a self-supervised feature extractor $f_{\theta_{\text{E}}}$ on images from the potentially poisoned dataset $\tilde D$.
> 2. Learn per-class densities $p_{\theta_{y}}(z)$ self-supervised features $z=f_{\theta_{\text{E}}}(x)$.
> 3. Identify disruptively and non-disruptively poisoned classes.
> 4. Assign poisoning scores $s_y(z)=\frac{p_{\theta_y}(z)}{\arg\max_{y'\neq y} p_{\theta_{y'}}(z)}$ to examples from classes $y$ identified as poisonous.
> 5. Split $\tilde D$ into subsets based on poisoning scores $s_y(z)$:
> 	* $\hat D_\text{P}$: poisoned examples,
> 	* $\hat D_\text{O}$: uncertain examples,
> 	* $\hat D_\text{C}$: clean examples from classes identified as targets and all examples from classes identified as clean.
> 6. Produce the cleansed poisoned subset $D_\text{P}'$ by assigning pseudo-labels to the examples from $\hat D_\text{P}$ according to $\arg\max_{y'\neq y}p_{\theta_{y'}}(z)$.
> 7. Produce the cleansed dataset as $\hat D' = \hat D_C \cup D_P'$.
> 8. Train a discriminative model on $\hat D'$.
>
> >Several typographical issues
>
> Thank you!
>
> >Table 4 should show the minimum poisoning rate where either the attacks or the defenses start to fail. For example, does your defense still work at 0.1% poisoning rate.
>
> This is a valuable suggestion. In our recent experiments,
> we discovered that a poisoning rate of 0.1% is indeed
> sufficient to compromise our defense. The scarcity of poisoned samples results in the failure to accurately identify the target class, consequently preventing the elimination of these poisoned samples.
>
> >Poisoning rate is also only one dimension of an attack's strength. Perturbation strength is an orthogonal dimension of attack strength against which the authors' defense is surely highly susceptible but is not explored in the empirical evaluation.
>
> We have noticed that some attacks that are non-disruptive become distruptive above some poisoning rate threshold.
> It also seems intuitive that stronger perturbations might make the attack more likely to be disruptive because the poisoned examples then bear less resemblance to their original class.
> In all our experiments, our defense has worked well in both cases.
>
> >The empirical evaluation's main results use either a 10% or 25% poisoning rate. In my view, those attack rates are wholly unrealistic for any marginally plausible real-world scenario. I would go so far as to consider those poisoning rates not meaningful to study since I cannot see a case where an attacker is inserting 25% poisoned data.
>
> We apologize if it was not clear: the 25% denotes that only 25% of the poisoned class consists of poisoned examples when the clean-label attack (LC) is used.
> We will run additional experiments with the clean-label attacks suggested by reviewer U1ob.
>
> We have considered attacks that poison 10% examples in order to be in line with previous work [1, 2].
> Do you have any suggestions about what to read to get a better idea of the attacker capabilities in important real-world scenarios?
>
> **Questions:**
>
> >At the beginning of Section 3.1, the authors write, "_We consider backdoor defenses that avoid standard supervised learning due to its sensitivity to poisoned labels and susceptibility to overfitting_." When I encountered this sentence when reading through the paper the first time, I did not understand what the authors meant, and after completing the paper, I am not sure I understood it.
>
> We aim for our defense to rely on self-supervised training, without labels.
> This approach is preferred because it is less likely to treat the trigger as a significant feature.
> In contrast, supervised training is likely to consider the trigger as the primary feature for identifying the label of poisoned examples.
> We will make this more clear in future revisions of our work.
>
> Please find the answers to your other comments in our response to all reviewers.
>
> [1] Li et al. - Neural Attention Distillation, https://arxiv.org/abs/2101.05930
> [2] Li et al. - Anti-Backdoor Learning, https://arxiv.org/abs/2110.11571

---

### Official Review · Reviewer_U1ob · 2023-10-30

**Soundness:** 3 good
**Presentation:** 2 fair
**Contribution:** 3 good
**Rating:** 5
**Confidence:** 3

**Summary:**

The paper proposes a novel method to detect and mitigate data poisoning attacks by means of per-class generative modeling. Instead of training the generative models on image space, which is claimed ineffective, the paper proposes to model the latent embeddings extracted by a self-supervised-learning feature extractor, using per-class normalizing flows. Then, it detects backdoor classes based on the average log-density over all foreign examples. Next, it computes a poisoning score to split samples of the identified target classes into clean, poisoned, and uncertain sets. Finally, the samples in the poisoned set are relabeled and combined with the clean set to produce a cleansed dataset for training. The method effectively mitigates common dirty-label attacks and one clean-label method on CIFAR-10, ImageNet, and VGGFace2.

**Strengths:**

- The paper explores a new approach for poisoning attack mitigation using generative modeling. While training the generative models on image space is ineffective, the paper proposes to use model the latent embeddings extracted by a self-supervised-learning feature extractor.
- The method effectively mitigates common dirty-label attacks and one clean-label method on CIFAR-10, ImageNet, and VGGFace2.

**Weaknesses:**

- The last two sentences in Section 4.5 are confusing. Although the numerator is extremely low, why is the score significantly higher? Also, if the poisoned samples score significantly higher than clean samples, aren't those poisoned samples mislabeled into the clean set? Finally, the arguments may be invalid by ignoring the change of the denominator.

- The paper experiments with only one clean-label attack, which injects adversarial noises into the poisoned data. Hence, it assumes that with clean-label attacks, the poisoned samples are completely distinct from the rest of the dataset in the self-supervised feature
space. That assumption may be incorrect with other clean-label attacks such as SIG [1] and Refool [2].

- The paper should discuss some adaptive attacks. For instance, the attacker can tune the backdoor trigger to fool a surrogate SimCLR model trained on clean data.

- The proposed method depends on too many hyper-parameters (\alpha, \lambda, \beta_{ND}, \beta_D). I cannot see how the selected values of these hyper-parameters are general and can work for all scenarios. The authors should ablate the choice of these hyper-parameters, particularly under different poisoning rates.

- \alpha is set as 0.15.
  - First, it means 70% of samples of the identified target class are uncertain and will be removed, which is a lot. It will weaken the cleansed dataset significantly, particularly when multiple (or all) classes are poisoned.
  - Also, all samples in the D_p are relabeled to a different class (Eq. 7), which is problematic if the percentage of the poisoned examples is less than 15% of the number of samples in the target class. For instance, in the case of CIFAR-10 with a 1% poisoning rate, the poisoned examples cover less than 9% of the samples in the target class, meaning more than 6% of the clean images in the target class are relabeled wrongly.
  - 15% of the samples in the target class are set as clean. It becomes problematic if the poisoned examples cover more than 85% of the samples of the target class. That situation can happen when the number of classes is high, e.g., a classification task with 100 classes and a poisoning rate of 10%.

- How does the algorithm behave in case the dataset is clean? And how does it behave under all2all attacks?

[1]. Mauro Barni, Kassem Kallas, and Benedetta Tondi. A new backdoor attack in cnns by training set corruption without label poisoning. In ICIP 2019.
[2]. Yunfei Liu, Xingjun Ma, James Bailey, and Feng Lu. Reflection backdoor: A natural backdoor attack on deep neural networks. In ECCV 2020.

**Questions:**

- The last two sentences in Section 4.5 are confusing. Although the numerator is extremely low, why is the score significantly higher? Also, if the poisoned samples score significantly higher than clean samples, aren't those poisoned samples mislabeled into the clean set? Finally, the arguments may be invalid by ignoring the change of the denominator.
- The authors should run the analysis in Fig. 1 and the experiments in Table 1 using SIG and Refool attacks?
- The authors should define and examine some potential adaptive attacks?
- The authors should ablate the choice of the hyper-parameters (\alpha, \lambda, \beta_{ND}, \beta_D), particularly under different poisoning rates.
- A more in-depth discussion on the impact of the value choice for \alpha.
- How does the algorithm behave in case the dataset is clean? And how does it behave under all2all attacks?

-

---

> ### Author Response · Authors · 2023-11-21
>
> Thank you for your feedback!
>
> >The last two sentences in Section 4.5 are confusing. Although the numerator is extremely low, why is the score significantly higher? Also, if the poisoned samples score significantly higher than clean samples, aren't those poisoned samples mislabeled into the clean set? Finally, the arguments may be invalid by ignoring the change of the denominator.
>
> We apologize. The penultimate sentence should have said *denominator* instead of *numerator*.
> The numerator will be higher because the generative model of the poisoned class is trained on the poisoned examples.
>
> If the above explanation does not resolve your doubts, please clarify "the change of the denominator".
>
> Please find the answers to your other comments in our response to all reviewers.

---

> ### Comment · Reviewer_U1ob · 2023-11-28
> **Thanks for the rebuttal. I decided to keep my initial score.**
>
> Thanks for your rebuttal and response. The above explanation resolve my doubts on the last two sentences in Section 4.5.
>
> I applaud the authors for their honest rebuttal. However, as pointed out, the paper has many problems that have to be addressed; thus, it is not ready for publication in its current form. For instance, the rebuttal confirms the problematic \alpha and the issue of the relabeling scheme. Although the authors had a chance to update their manuscript, some issues have not been adequately addressed, including the sensitivity of their approach to the hyperparameters, adaptive attacks, and performance under all2all attacks. Also, I agree with the concern from Reviewer gUtc about expensive computation costs.
>
> Hence, I decided to keep my initial score.
>
> Best regards,
> Reviewer U1ob

---

### Official Review · Reviewer_gUtc · 2023-11-02

**Soundness:** 3 good
**Presentation:** 2 fair
**Contribution:** 3 good
**Rating:** 5
**Confidence:** 4

**Summary:**

This paper proposes a new method for robust training of neural network classifiers against backdoor data poisonings. In short, backdoor attacks aim to create hidden associations between a trigger and a target class by poisoning a small portion of the training data. This can be done via attaching small triggers to the image and optionally changing the labels associated with each image. If the label is changed, the attack is called poisoned-label attacks, while attacks that do not change the labels are called clean label attacks.

In this paper, the authors proposes a three stage process for purifying the poisoned dataset and training a robust model that is free of backdoors. In the first step, they use a self-supervised method such as SimCLR to train a feature extractor on the poisoned dataset. Using this feature extractor, they then get the feature representations of all the training samples. In the second step, they train a generative model (here normalizing flows) for each class representation. Using these normalizing flows, they then defined a likelihood-based score function to identify poisonous samples from clean ones. This step is motivated by earlier observations on the feature space representation of backdoor attacks using self-supervised models. In particular, for samples that are likely to be poisoned-label attacks, the proposed method can identify the target class and correctify their labels. Some samples are also removed from the training dataset if they do not belong to any of the previous categories. Once this step is done, a neural network is finally trained over the purified dataset.

Experimental results over CIFAR-10, ImageNet-30, and VGGFace indicate the effectiveness of the proposed method against poisoned-label and clean-label backdoor attacks.

**Strengths:**

- The proposed method is novel and interesting. It is based on an empirical observation around the feature space representation of poisoned data in the feature space of models trained with self-supervised learning. The use of normalizing flows to model the per-class latent space distribution is also novel.

- Empirical results indicate that this three stage method can mitigate the effect of backdoor attacks. Even more interestingly, it can revive the poisoned-label samples and re-use them in the training process.

**Weaknesses:**

- Even though the proposed method is working well, it is highly inefficient and requires lots of compute. In particular, the proposed method starts with self-supervised pre-training of a feature extractor using SimCLR. Then, it trains _one_ normalizing flow _per each class_ to finally be able to get rid of poisonous samples and start training a robust model. Such extensive use of resources is quite intensive, and frankly speaking, might be redundant. The field of backdoor defense has came up with alternative solutions such as [1-4] that are far more efficient than the proposed solution where some of them just take one training round to give robust models. Apart from the expensive self-supervised training at the beginning, the proposed solution requires one flow-based model per class which means that its resources grows linearly with the number of classes.

- In lieu of the previous issue, first the paper needs to include more recent baselines [1-3], and second, it is important to include the total training time (from start to delivering a robust model) for all of the methods. This way, the readers can have a better understanding of the computational efficiency of current methods.

- There are certain parts in the paper that might cause confusion. For instance, the explanations given in Sections 4.5-4.6 are seem contrasting. On the one hand, the paper says that for clean samples the score $s\_{y}(\boldsymbol{z})$ is higher. On the other hand, the same score is also higher for disruptive poisoning. Figure 4 also shows the same trend for both the clean samples as well as disruptive attacks. I think that these two sections should be re-written (see below for questions), because currently it seems that some of the clean samples can also initially be removed by this method. If this is the case, it should be explained. Optionally, adding a diagram of step-by-step poisoned sample removal might also be helpful.

[1] Liu, Min, et al. "Beating Backdoor Attack at Its Own Game." _ICCV_, 2023.

[2] Huang, Hanxun, et al. "Distilling Cognitive Backdoor Patterns within an Image." _ICLR_, 2023.

[3] Dolatabadi, Hadi, et al. "Collider: A robust training framework for backdoor data." _ACCV_, 2022.

[4] Hayase, Jonathan, et al. "Spectre: Defending against backdoor attacks using robust statistics." _ICML_, 2021.

**Questions:**

- Claiming that this work is "the first backdoor defense based on generative modelling" is inaccurate. For one, MESA [5] has also used generative modelling as a solution to neural backdoors.

- Why do we need to identify/remove poisonous samples using class conditional normalizing flows and then train another neural network of our task? In other words, can't we just use the per-class normalizing flow for classification as well? Running experiments on this scenario is highly encouraged.

- Can you repeat the same process for generating Figure 1 for other attacks?

- Based on the Figure 1 (left), the proposed method heavily relies on the fact that the poisoned samples in the target class are scarce. What happens if the number of poisoned samples (those with triggers) that use the same trigger are abundant? Experiments on this scenario is highly encouraged.

- Section 4.5 and 4.6 are rather confusing. Can you please elaborate on the filtration procedure? The paper currently says that "We include the samples with $\alpha$ highest poisoning scores in $\hat{\mathcal{D}}\_{\mathrm{P}}$, and include the samples with the $\alpha$ lowest poisoning scores together with samples from identified clean classes in $\hat{\mathcal{D}}\_{\mathrm{C}}$." Do these two steps done on the same score graph? Does this mean that some of the clean samples are also removed? Potentially, is this the reason for the under-performance of the proposed method in the case of high poison rate (Table 4)?

- What experimental settings (number of epochs, etc.) are used for SimCLR? What is the architecture of normalizing flows?

- As mentioned above, add the mentioned baselines and report the total training time for all of the methods to see the computational efficiency.

- Why so many number of epochs (200) is used for training models? Usually, 120 epochs is enough to train ResNet models with SGD on CIFAR-10.

[5] Qiao, Ximing, Yukun Yang, and Hai Li. "Defending neural backdoors via generative distribution modeling." _NeurIPS_, 2019.

---

> ### Author Response · Authors · 2023-11-21
>
> Thank you for your feedback!
>
> **Weaknesses**
>
> In case of a dataset with input size $224\times224$, 10 classes and ~5000 example per class,
> self-supervised representation learning takes 10000 s,
> supervised training (200 epochs) takes 2750 s,
> while the training of generative classifiers takes about 90 s per class regardless of image size.
> We observe that our generative setups involve a neglectable processing overhead.
> Moreover, this can be further improved by replacing C per-class models with 1 model of joint density $p[z, y]$.
> Self-supervised learning indeed is computationally expensive, however that seems as an acceptable price for a substantial reduction of vulnerability to cybernetic attacks.
>
> We will try to address the last point by improving the explanations.
>
> **Questions**
>
> >Why do we need to identify/remove poisonous samples using class conditional normalizing flows and then train another neural network of our task? In other words, can't we just use the per-class normalizing flow for classification as well? Running experiments on this scenario is highly encouraged.
>
> We agree that it would make sense to test the normalizing flows as the final classifier or test such a classifier re-trained on the cleansed dataset.
> Given our assumptions, we expect the following:
> - In case of non-disruptive attacks, we expect such a classifier to correctly classify some, but still be ambiguous and misclassify many of the poisoned examples with respect to the original class. In cases when it is ambiguous on two classes, those classes might often be the original (poisoned) class and the target class.
> - In case of disruptive attacks, we would expect such a classifier to often classify the poisoned examples into the target class because they are not similar to their original class in the self-supervised feature space.
> We will test this empirically.
>
> Most practical setups will prefer to re-train the discriminative model on cleansed data since i) discriminative models perform better on discriminative tasks.
> Note that the resulting computational overhead is not large compared to feature extraction, and that it would be an acceptable price for the delivered cleansing performance. Recent comparison of generative and discriminative classification performance can be found in [1].
>
>
> >Can you repeat the same process for generating Figure 1 for other attacks?
>
> Yes. We will generate Figure 1 for other attacks and include it in the future revisions of our work.
>
> >What experimental settings (number of epochs, etc.) are used for SimCLR? What is the architecture of normalizing flows?
>
> We perform SimCLR training for 100 epochs with a batch size of 256 and
> Adam optimizer with
> a fixed learning rate of $3\cdot10^{-4}$.
> Our per-class normalizing flow consists of two steps which consist of actnorm [4] and an affine coupling [5].
> Each coupling module computes the affine parameters with a pair of fully-connected layers with ReLU between them.
> We shall include these pieces of information in the next revision of our work.
>
> >Why so many number of epochs (200) is used for training models? Usually, 120 epochs is enough to train ResNet models with SGD on CIFAR-10.
>
> We were following the training configurations from previous work [2, 3].
>
> Please find the answers to your other comments in our response to all reviewers.
>
> [1] Mackowiak et al. - Generative Classifiers as a Basis for Trustworthy Image Classification, https://arxiv.org/abs/2007.15036
> [2] Huang et al. - Backdoor Defense via Decoupling the Training Process, https://arxiv.org/abs/2202.03423
> [3] Gao et al. - Backdoor Defense via Adaptively Splitting Poisoned Dataset, https://arxiv.org/abs/2303.12993
> [4] Kingma et al. - Glow: Generative Flow with Invertible 1x1 Convolutions, https://arxiv.org/abs/1807.03039
> [5] Dinh et al. - Density estimation using Real NVP, https://arxiv.org/abs/1605.08803

---

> > ### Comment · Reviewer_gUtc · 2023-11-23
> > **Reply to Authors**
> >
> > I want to thank the authors for their response. Unfortunately, the authors' response did not address my concerns that I mentioned in my initial review. In particular, additional comparisons with newer existing baselines in terms of both the attack success rate as well as total wall-time is required to demonstrate whether the proposed method is making an efficient use of resources or not. Also, as evident in Table 1, the proposed method is getting rid of some clean samples and the benign accuracy (ACC) is usually lower than that of other methods. To evade this, the paper has use the notion of (ACC-ASR) to determine which method is most successful, but in my view, both ACC and ASR should be considered separately. The issue of filtering clean data might get even worse as we increase the filtration threshold which in turn can hurt the clean accuracy.
> >
> > Overall, I believe that my initial assessment is a fair evaluation of this paper. That being said, I am open to discussion, especially from my peers.

---

### Official Review · Reviewer_rmHx · 2023-11-07

**Soundness:** 2 fair
**Presentation:** 2 fair
**Contribution:** 2 fair
**Rating:** 5
**Confidence:** 3

**Summary:**

This paper proposes to model the per-class distribution with a generative model and uses it to sanitise the data against backdoors. The approach operates in a latent dimension as opposed to the input space. Under the assumption and empirical observation (Figure 3, Figure 4) that the poisoned samples will exhibit different density scores for their target classes, they use a threshold to identify the poisoned samples in two scenarios

**Strengths:**

- This paper builds on an empirical observation that comparing per-class densities over extracted features can reveal poisoning behaviour.

- The ablation study shows that choice of feature representation doesn’t significantly vary the performance with two models resulting in marginal differences in attack success rate and accuracy

**Weaknesses:**

- I think the paper can improve with investigation of what makes a generative model stand out? Perhaps an investigation of how adaptive attacks can circumvent the threshold based detector?

- Similarly, the paper can also benefit from investigation of the choice of threshold \beta_ND and \beta_D.

- Figure 1 and its legend are a bit small to read

**Questions:**

- How sensitive is this method with respect to choice of thresholds?

- It is unclear what the limits of this approach are? Are there scenarios where this detector will fail to detect?

---

> ### Author Response · Authors · 2023-11-21
>
> Thank you for your feedback!
>
> >I think the paper can improve with investigation of what makes a generative model stand out?
>
> If we understand correctly, we agree that we should test how important the conditional generative model $p[z\mid y]$ is by comparing its feature densities with class probabilities of a suitable discriminative model $p[y\mid z]$.
>
> Please find the answers to your other comments in our response to all reviewers.

---

### Author Response · Authors · 2023-11-21
**Response to all reviewers**

We thank the reviewers for their feedback.

We find the feedback to be quite valuable, and agree that there are some important things that we have to address for our work to be ready for publication.

Here are our responses to some observations and questions that are shared by multiple reviewers.

---

> ### Author Response · Authors · 2023-11-21
> **Effect of large poisoning rates**
>
> Reviewer U1ob (R3):
> >alpha is set as 0.15.
> >- [...]
> >- 15% of the samples in the target class are set as clean. It becomes problematic if the poisoned examples cover more than 85% of the samples of the target class. That situation can happen when the number of classes is high, e.g., a classification task with 100 classes and a poisoning rate of 10%.
>
> These are some interesting observations, especially the last point.
> We have conducted an experiment on the CIFAR-100 dataset using a poisoning rate of 10%.
> As you pointed out, with $\alpha=0.15$, a significant
> section of poisoned samples is identified as clean, making our defense ineffective for this scenario.
> By setting $\alpha=0.05$, we succeeded in reducing ASR while preserving the accuracy of the clean model.
> We note that even in this case there are some poisoned samples left in the purified dataset.
> However, their impact was too small to successfully implement the backdoor.
> In future revisions of our work, we will investigate the selection of $\alpha$.
>
> You are also right about relabeling of clean examples. This is also related to the effect of accuracy on clean data *decreasing*
> as the poisoning rate *decreases* in Table 4.
> We will comment on this in the paper, and also include some experiments with no relabeling.
>
> Reviewer gUtc (R2):
> >Based on the Figure 1 (left), the proposed method heavily relies on the fact that the poisoned samples in the target class are scarce. What happens if the number of poisoned samples (those with triggers) that use the same trigger are abundant? Experiments on this scenario is highly encouraged.
>
> If we understand well, this is addressed above.
>
> >The paper currently says that "We include the samples with highest poisoning scores in $\hat{D}\_{\text{P}}$, and include the samples with the $\alpha$ lowest poisoning scores together with samples from identified clean classes in $\hat{\mathcal{D}}_{\mathrm{C}}$."Do these two steps done on the same score graph? Does this mean that some of the clean samples are also removed?
>
> The filtering is done on each poisoned class separately.
> We put the $\alpha$ highest scoring examples from each class that is identified as poisoned (target) into $\hat{\mathcal{D}}\_\text{P}$.
> The $\alpha$ lowest scoring examples from these classes and **all** examples from classes identified as clean go to $\hat{\mathcal{D}}_\text{C}$.
> Clean examples can be removed or relabeled only if they are from classes identified as poisoned.
> We will clarify this in the revised paper.
>
> >Potentially, is this the reason for the under-performance of the proposed method in the case of high poison rate (Table 4)?
>
> In Table 4, $\alpha=0.15$ in all experiments. Our best guess is that, as the poisoning rate increases,
> - ASR increases because more poisoned examples are misidentified as clean (they pass the $1-\alpha$ threshold), and
> - accuracy increases because a smaller number of clean examples ends up in $\hat{\mathcal D}_\text{P}$ and is incorrectly relabeled.

---

> ### Author Response · Authors · 2023-11-21
> **Hyperparameter validation**
>
> Reviewer rmHx (R1):
> >Similarly, the paper can also benefit from investigation of the choice of threshold \beta_ND and \beta_D.
>
> Reviewer U1ob (R3):
> >The proposed method depends on too many hyper-parameters (\alpha, \lambda, \beta_{ND}, \beta_D). I cannot see how the selected values of these hyper-parameters are general and can work for all scenarios. The authors should ablate the choice of these hyper-parameters, particularly under different poisoning rates.
>
> The next revision of our work will report the sensitivity of our approach
> to hyperparameters $\alpha$, $\lambda$, $\beta_\text{ND}$, $\beta_\text{D}$
> and propose an adaptive selection of $\alpha$.

---

> ### Author Response · Authors · 2023-11-21
> **Adaptive attacks**
>
> Reviewer U1ob (R3):
> >The paper should discuss some adaptive attacks. For instance, the attacker can tune the backdoor trigger to fool a surrogate SimCLR model trained on clean data.
>
> Reviewer y1uP (R4):
> >Like all other empirical defenses, the authors' method comes with no guarantees on its effectiveness. In my view, empirical defense papers must meet two necessary conditions to be fit for publication. Unfortunately, this paper does not meet either.
>     1. The paper should explicitly note that their method comes with no guarantees and contrast this weakness against the plethora of papers (e.g., [1]) that provide methods certifiably robust to training data attacks.
>     2. The paper should ideally evaluate against an adaptive attacker who is aware of their defense and actively tries to avoid it. At the very minimum, the paper should include a convincing discussion of why an adaptive attacker is not feasible or reasonable.
>
> We plan to evaluate the resilience of our defense against adaptive attacks.
> An example of an adaptive attack that might be able to breach our defense involves optimizing the trigger such that it introduces
> similarities between poisoned examples containing the trigger and clean examples of the target class in the self-supervised latent space [1].
> We will include these experiments in the next revisions of our work.

---

> ### Author Response · Authors · 2023-11-21
> **Contribution claims**
>
> Reviewer gUtc (R2):
> >Claiming that this work is "the first backdoor defense based on generative modelling" is inaccurate. For one, MESA [5] has also used generative modelling as a solution to neural backdoors.
>
> Reviewer y1uP (R4):
> >On page 2, you summarize the paper's second contribution writing, "_We propose the first backdoor defense based on generative modeling_." This is a very broad claim that I suspect is not true.
>
> You are right. We will address this in our revision. The MESA [4] approach models a distribution of triggers, which aids in a strong backdoor defense, operating under the assumption that the approximate size of the trigger is known.
> In contrast, our method models the distribution
> of the training dataset (in the latent space) without relying on any assumptions such as trigger size or additional clean data.

---

> ### Author Response · Authors · 2023-11-21
> **Weaknesses of the defense**
>
> Reviewer rmHx (R1):
> We agree that we should conduct better theoretical and empirical investigations on the weaknesses of the proposed method.
> We will include a section about limitations and improve the figures.
>
> Reviewer U1ob (R3):
> >The paper experiments with only one clean-label attack, which injects adversarial noises into the poisoned data. Hence, it assumes that with clean-label attacks, the poisoned samples are completely distinct from the rest of the dataset in the self-supervised feature space. That assumption may be incorrect with other clean-label attacks such as SIG and Refool.
>
> For now, we have tested the SIG attack following the setups from previous work [2].
> Similar to Label Consistent attack [3],
> our method classified it as the disruptive attack. After retraining, the ASR was brought down to 0.00% while keeping high accuracy on clean data of 93.2%.
> We will also look into the Refool paper and update our work accordingly.
>
> Reviewer U1ob (R3):
> >How does the algorithm behave in case the dataset is clean?
>
> When we evaluate on clean datasets, no poisoned class is detected and the result is the same as if no defense is used.
> Nonetheless, this does not assure that the detection method will always be free from false positives due to the usage of a specific threshold.
> In case that a class is misidentified as poisonous, accuracy will decrease because of relabeling and filtering like in the rows with low poisoning rates in Table 4.
>
> > And how does it behave under all2all attacks?
>
> Some possibilities for label-modifying attacks are:
> 1. one class is poisoned with examples originally from all other classes (a single trigger).
> 2. one class is poisoned with examples originally from one other class (a single trigger),
> 3. all classes $i$ are poisoned with examples originally from the class $(i+1) \bmod K$ (a single trigger),
> 4. all classes are poisoned with examples from all other classes (one trigger for each target).
> Following previous work, our experiments use the first case.
> Now we have also performed experiments with the third case, and notice that there are some false negatives in the detection of poisoned classes.
> In this case, this is because examples from all classes except the source class are contributing low log-densities to $S_\text{ND}^y$.
> We expect similar results in the second case.
> For the fourth case, we expect poisoned class detection performance similar to that in the first case.
> However, it is not clear whether this case is realistic, since we have not seen it in the literature.

---

> ### Author Response · Authors · 2023-11-21
> **References**
>
> [1] Huang et al. - Backdoor Defense via Decoupling the Training Process, https://arxiv.org/abs/2202.03423
> [2] Li et al. - Anti-Backdoor Learning: Training Clean Models on Poisoned Data, https://arxiv.org/abs/2110.11571
> [3] Turner et al. - Label-Consistent Backdoor Attacks, https://arxiv.org/abs/1912.02771
> [4] Qiao et al. - Defending Neural Backdoors via Generative Distribution Modeling, https://proceedings.neurips.cc/paper_files/paper/2019/file/78211247db84d96acf4e00092a7fba80-Paper.pdf

---

### Meta-Review · Area_Chair_M72Z · 2023-12-06

**Metareview:**

This paper proposes to model the per-class distribution with a generative model and uses it to purify the data against backdoors. However, the paper still has a lot of problems like hyper-parameter sensitivity, computation cost, and others pointed out by reviewers. Thereby, I agree with all the reviewer's decision to reject this paper. Hope the authors can modify their work in the revision according to the reviewers' advice.

**Justification For Why Not Higher Score:**

There are a lot of problems pointed out by reviewers.

**Justification For Why Not Lower Score:**

N/A

---

### Decision · Program_Chairs · 2024-01-16

Reject